# Association between COPD and Stage of Lung Cancer Diagnosis: A Population-Based Study

Stacey J. Butler [1,2,3] , Alexander V. Louie [3], Rinku Sutradhar [2,3,4], Lawrence Paszat [2,3,5], Dina Brooks [6] and Andrea S. Gershon [1,2,3,5,*]

1    Institute of Medical Sciences, University of Toronto, Toronto, ON M5S 1A8, Canada; stacey.butler@utoronto.ca
2    ICES, Toronto, ON M4N 3M5, Canada; rinku.sutradhar@ices.on.ca (R.S.);
     lawrence.paszat@sunnybrook.ca (L.P.)
3    Sunnybrook Research Institute, Sunnybrook Health Sciences Centre, Toronto, ON M4N 3M5, Canada;
     alexander.louie@sunnybrook.ca
4    Dalla Lana School of Public Health, University of Toronto, Toronto, ON M5T 3M7, Canada
5    Institute of Health Policy, Management and Evaluation, University of Toronto, Toronto, ON M5T 3M6, Canada
6    School of Rehabilitation Sciences, McMaster University, Hamilton, ON L8S 1C7, Canada;
     brookd8@mcmaster.ca
*    Correspondence: andrea.gershon@ices.on.ca

**Abstract:** Chronic obstructive pulmonary disease (COPD) is associated with an increased risk of lung cancer; however, the association between COPD and stage of lung cancer diagnosis is unclear. We conducted a population-based cross-sectional analysis of lung cancer patients (2008–2020) in Ontario, Canada. Using estimated propensity scores and inverse probability weighting, logistic regression models were developed to assess the association between COPD and lung cancer stage at diagnosis (early: I/II, advanced: III/IV), accounting for prior chest imaging. We further examined associations in subgroups with previously diagnosed and undiagnosed COPD. Over half (55%) of all lung cancer patients in Ontario had coexisting COPD (previously diagnosed: 45%, undiagnosed at time of cancer diagnosis: 10%). Compared to people without COPD, people with COPD had 30% lower odds of being diagnosed with lung cancer in the advanced stages (OR = 0.70, 95% CI: 0.68 to 0.72). Prior chest imaging only slightly attenuated this association (OR = 0.77, 95% CI: 0.75 to 0.80). The association with lower odds of advanced-stage diagnosis remained, regardless of whether COPD was previously diagnosed (OR = 0.68, 95% CI: 0.66 to 0.70) or undiagnosed (OR = 0.77, 95% CI: 0.73 to 0.82). Although most lung cancers are detected in the advanced stages, underlying COPD was associated with early-stage detection. Lung cancer diagnostics may benefit from enhanced partnership with COPD healthcare providers.

**Keywords:** lung cancer; diagnostics; chronic obstructive pulmonary disease

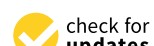



## 1. Introduction

Given the shared risk factors and underlying disease mechanisms, chronic obstructive pulmonary disease (COPD) is a common comorbidity among lung cancer patients [1]. People with COPD have a higher risk of developing lung cancer compared to people without COPD, regardless of their smoking history [2–4]. People with COPD also have worse lung cancer outcomes when compared to people without COPD [5,6] and utilize more healthcare resources [6,7]. Diagnosis of lung cancer in the early stages is imperative to initiate cancer treatment and give patients the best chance at survival. Unfortunately, approximately half of all lung cancers are diagnosed in stage IV, when treatment options are limited and survival is very poor [8,9]. There are several patient-, disease-, and health-system-related factors that can facilitate or delay the diagnosis of lung cancer [10,11]; however, little attention has been given to comorbid respiratory diseases such as COPD.

Lung cancer may be detected in the early stages among people with COPD due to established healthcare access and routine assessment of respiratory symptoms, but this has not been confirmed. Early diagnosis of lung cancer among patients with COPD would allow for earlier intervention, treatment, and improved prognosis. On the contrary, advanced-stage diagnosis of lung cancer in patients with COPD would place them at further risk of worse outcomes. There are several reasons why lung cancers could be detected in the advanced stages among individuals with COPD. A recent qualitative study by Cunningham et al. [12] revealed that people with COPD were frequently unaware of their increased lung cancer risk, and changes or worsening in symptoms were often attributed to their COPD, without consideration of lung cancer. Two small studies also suggest that prior respiratory diseases can mask the symptoms of lung cancer and delay diagnosis [13,14]. Additionally, COPD itself is widely underdiagnosed, with global estimates suggesting that up to 81% of people with COPD have undiagnosed disease [15]. People with undiagnosed COPD would be at further risk of detecting lung cancer in the advanced stages. This population includes nonsmokers, where COPD or lung cancer may not be suspected, as well as people with a smoking history who often delay seeking care due to worry, embarrassment, self-blame, or perceived stigma from healthcare providers [11,16–18].

It is not currently known whether people with COPD are more likely to be diagnosed with lung cancer in the early or advanced stages. Our population-based study sought to fill in this knowledge gap. Our aim was to determine whether COPD is associated with diagnosis of lung cancer in the early (I/II) or advanced stages (III/IV). We also considered the impact of previously diagnosed COPD and undiagnosed COPD, while accounting for other variables which are associated with lung cancer diagnostic stage, including prior chest imaging, established healthcare access, comorbidities, and sociodemographic factors.

## 2. Materials and Methods

### 2.1. Study Design and Setting

We conducted a population-based, retrospective cross-sectional study using linked Ontario health administrative databases and cancer registries available through ICES, in Toronto, Ontario, Canada. ICES is an independent, nonprofit research institute whose legal status under Ontario's health information privacy law allows it to collect and analyze health care and demographic data, without consent, for health system evaluation and improvement. This project was approved by the University of Toronto Health Sciences Research Ethics Board.

### 2.2. Data Sources

Ontario has a universal healthcare program where residents have public health insurance under the Ontario Health Insurance Plan (OHIP), which covers all medically necessary services provided by physicians and hospitals in the province, with details captured in health administrative databases. These databases cover virtually all outpatient, emergency department, and hospital-based care. The Ontario Cancer Registry (OCR) provides population-based data on cancers diagnosed in Ontario and disease characteristics. These datasets were linked using unique encoded identifiers and analyzed at ICES. Additional information on databases used in this study is available through the ICES Data Dictionary [19].

### 2.3. Study Population

We included all Ontario residents, aged 18 and over, diagnosed with primary lung cancer (International Classification of Diseases for Oncology (ICD-O) code C34, morphology codes available in Supplemental Table S1) between 1 April 2008 and 1 January 2020. We excluded individuals with stage 0 or occult lung cancer, missing or unknown stage, and missing covariate information. The index date was set as the date of lung cancer diagnosis in the OCR.

### 2.4. Primary Outcome

The primary outcome was the stage of lung cancer at diagnosis, defined as early (stage I/II) or advanced (stage III/IV). We utilized the "Best Stage" definition from the OCR, which is a collaborative staging method that combines the clinical and pathological data to determine a modified (or adapted) TNM stage.

### 2.5. Exposures

The primary exposure was underlying/co-existing COPD, which was ascertained using a previously validated case definition of at least one ambulatory claim or hospitalization for COPD based on ICD-9 codes 491, 492, or 496 and ICD-10 codes J41, J42, J43, or J44 [20]. This case definition has been shown to have 85% sensitivity and 78% specificity compared to a clinical reference standard [20] and was used previously to study COPD in Ontario [21,22]. Individuals were considered to have "previously diagnosed COPD" if they met this case definition prior to the time of lung cancer diagnosis, defined as >90 days prior to the diagnosis date. Individuals with COPD diagnosed near the time of lung cancer diagnosis (+/− 90 days) were considered to have "undiagnosed COPD".

To account for prior imaging, we considered having prior chest computed tomography (CT) scans as a secondary exposure. We excluded any chest CT scans that occurred during the 90 days prior to lung cancer diagnosis (i.e., during the "diagnostic period") as these were likely related to the diagnosis of lung cancer itself. Individuals were classified as having prior imaging if they had any chest CT scan in the five years prior to the diagnostic period.

### 2.6. Covariates

Canadian Census definitions were used to categorize individuals as residing in rural (<10,000 residents) or urban areas. Neighbourhood income quintiles were used as a measure of socioeconomic status for urban areas. The Immigration, Refugees and Citizenship Canada (IRCC) database captures immigration to Canada since 1985. Immigration status was also considered and patients were categorized as recent immigrants to Canada (≤10 years), long-term residents (>10 years), or nonimmigrants since 1985.

Comorbidities were ascertained using validated algorithms to identify individuals with asthma [23] (since April 1996), dementia (since 1966) [24], diabetes [25] (since April 1991), or congestive heart failure (CHF) [26] (since April 1991). Pneumonia in the five years prior to index was ascertained using ICD-10 codes "J10–18" and ICD-9 codes "480–487" for outpatient, emergency department visits or hospitalizations. We also considered whether individuals were under surveillance for a previous cancer (excluding previous lung cancer) in the five years prior to index.

Healthcare visits during the 90 days prior to lung cancer diagnosis were not considered, as these were more likely to be associated with the diagnosis of lung cancer and its workup, and not reflective of healthcare utilization prior to the development of lung cancer. This 90-day period prior to lung cancer diagnosis was referred to as the "diagnostic period". Prior healthcare utilization was assessed by the rate of outpatient primary care visits in the five years prior to the 90-day diagnostic period. Whether the patient was under specialist care, considered as a visit to a respirologist, cardiologist, or internal medicine specialist, in the five years prior to the 90-day diagnostic period was considered as a binary covariate (yes/no). These specialists were included because they are primarily involved in caring for patients with COPD and related comorbidities [22].

### 2.7. Statistical Analyses

We utilized propensity score methods (estimating the probability of having COPD) and inverse probability weighting (IPW) [27] to determine whether COPD status was associated with the diagnostic stage of lung cancer. This method allows us to estimate the effect of COPD on the stage of lung cancer using observational data and minimize potential confounding bias [27]. To determine the association between COPD and stage of lung cancer, we estimated propensity scores (the probability of a patient having COPD

given their characteristics). Propensity scores were transformed to weights to estimate the average treatment effect (ATE) [27,28] and then stabilized to ensure proper estimation of the variance and reduce sample size inflation [28]. Balance was assessed in the distributions of prespecified covariates (listed above) among the COPD exposure groups using weighted regression and calculating absolute standardized mean differences (SMD) of the effect size [29]. An SMD < 0.1 was considered sufficiently balanced.

Unweighted and weighted logistic regression analyses were used to determine whether COPD status was associated with early or advanced-stage lung cancer. To address the risk of stage attributable to prior imaging, we performed additional analyses which adjusted the weighted multivariable model for the secondary exposure (any chest CT scan). The regression results are presented as odds ratios (ORs) with 95% confidence intervals (CI).

An a priori subgroup analysis was performed within the COPD cohort to address the role of previously diagnosed COPD and undiagnosed COPD. Propensity scores and stabilized weights were re-estimated for all subgroup analyses using a binary COPD exposure variable, and covariate balance was reassessed as per the methods described above. Balance assessments for the subgroup analyses are provided in Supplemental Tables S2–S4. All statistical analyses were conducted using R statistical software 3.1.2 [30] with a significance level of 0.05.

### *2.8. Sensitivity Analyses*

A sensitivity analysis with re-estimation of the propensity scores was performed to account for possible misclassification of COPD. Among the subgroup with previously diagnosed COPD, a stricter definition for COPD (two or more outpatient claims or at least one hospitalization for COPD within a two-year period prior to lung cancer diagnosis) with higher specificity (91.5%) was used [20]. Individuals with previously diagnosed COPD who did not meet these criteria were considered to not have COPD in the sensitivity analysis.

### 3. Results

A total of 103,395 individuals were diagnosed with lung cancer in Ontario during the study period, with 86,835 meeting the inclusion criteria (Figure 1). Among those who were excluded due to missing stage (15.7%), over half (54.5%) had COPD (previously diagnosed—48.0%, undiagnosed—6.5%). An additional 156 individuals (0.2%) were excluded due to stage 0 or occult lung cancer and 204 individuals were excluded (0.2%) due to missing covariate information.

The baseline characteristics of the study cohort are provided in Table 1. In total, 55.0% of lung cancer patients had coexisting COPD, which was further stratified as previously diagnosed COPD (81.5%) and undiagnosed COPD (18.5%). For those with previously diagnosed COPD (*n* = 38,894), the median time between COPD and lung cancer diagnosis was 10 years (IQR: 5 to 16 years). Among the individuals with undiagnosed COPD (*n* = 8843), 16% were diagnosed with both COPD and lung cancer on the same day and most COPD diagnoses (58%) occurred within 30 days of lung cancer diagnosis.

**Table 1.** Lung cancer cohort characteristics by COPD status (*N* = 86,365).

| | COPD | | | |
|---|---|---|---|---|
| | **Total COPD** | **Previously Diagnosed COPD** | **Undiagnosed COPD** | **No COPD** |
| **Characteristics** | (*n* = 47,737) | (*n* = 38,894) | (*n* = 8843) | (*n* = 39,098) |
| **Lung Cancer Stage** | | | | |
| Early | 31.1% | 32.0% | 27.2% | 23.9% |
| Advanced | 68.9% | 68.0% | 72.8% | 76.1% |
| **Type of Lung Cancer** | | | | |
| NSCLC: Adenocarcinoma | 36.7% | 35.5% | 42.2% | 49.4% |
| NSCLC: Squamous cell carcinoma | 21.7% | 21.9% | 20.5% | 13.9% |
| NSCLC: Large cell carcinoma | 2.5% | 2.4% | 2.6% | 2.7% |
| NSCLC: Adenocarcinoma | 0.6% | 0.5% | 0.7% | 0.6% |
| NSCLC: Not otherwise specified | 14.1% | 14.2% | 13.6% | 14.2% |
| SCLC | 12.5% | 12.7% | 11.7% | 10.4% |
| Unspecified | 12.0% | 12.7% | 8.8% | 8.8% |
| **Method of Confirmation** | | | | |
| Clinical | 1.3% | 1.4% | 1.0% | 0.8% |
| Histology/Cytology | 87.1% | 85.9% | 92.2% | 91.1% |
| Imaging | 4.4% | 5.0% | 2.0% | 2.7% |
| Unknown | 7.2% | 7.8% | 4.8% | 5.4% |
| Age, mean (SD) | 71.3 (9.6) | 71.9 (9.4) | 68.7 (9.9) | 68.9 (11.3) |
| Sex, % male | 52.1% | 51.5% | 54.7% | 51.0% |
| **Rurality/Income Quintile** | | | | |
| Rural | 16.3% | 16.5% | 15.5% | 13.8% |
| Urban 1 (lowest) | 22.9% | 23.3% | 21.3% | 18.2% |
| Urban 2 | 19.7% | 19.6% | 20.3% | 18.8% |
| Urban 3 | 15.8% | 15.8% | 15.7% | 17.1% |
| Urban 4 | 13.6% | 13.4% | 14.4% | 16.5% |
| Urban 5 (highest) | 11.6% | 11.4% | 12.8% | 15.6% |
| **Immigration Category** | | | | |
| ≤10 years | 0.6% | 0.4% | 1.5% | 2.2% |
| Long-term resident (>10 years) | 2.9% | 2.7% | 4.2% | 6.6% |
| Nonimmigrant | 96.5% | 97.0% | 94.4% | 91.2% |
| **Comorbidities** | | | | |
| Asthma | 23.1% | 26.1% | 9.9% | 7.9% |
| Congestive heart failure | 16.2% | 17.7% | 9.7% | 7.5% |
| Dementia | 5.0% | 5.5% | 2.8% | 3.7% |
| Diabetes | 28.3% | 29.4% | 23.3% | 24.8% |
| Previous pneumonia | 30.9% | 32.5% | 24.3% | 18.5% |
| Cancer in the previous 5 years | 8.3% | 8.7% | 6.6% | 9.1% |
| **Rate of Primary Care Visits** | | | | |
| ≥5 per year | 47.0% | 50.6% | 31.3% | 34.0% |
| >2 and <5 per year | 35.4% | 35.1% | 37.0% | 38.1% |
| >0 and ≤2 per year | 14.9% | 12.6% | 24.9% | 22.3% |
| 0 | 2.6% | 1.7% | 6.7% | 5.6% |
| Specialist care | 46.9% | 49.7% | 34.7% | 36.4% |

Abbreviations: NSCLC = non-small cell lung cancer, SCLC = small cell lung cancer, SMD = standardized mean difference, SD = standard deviation.

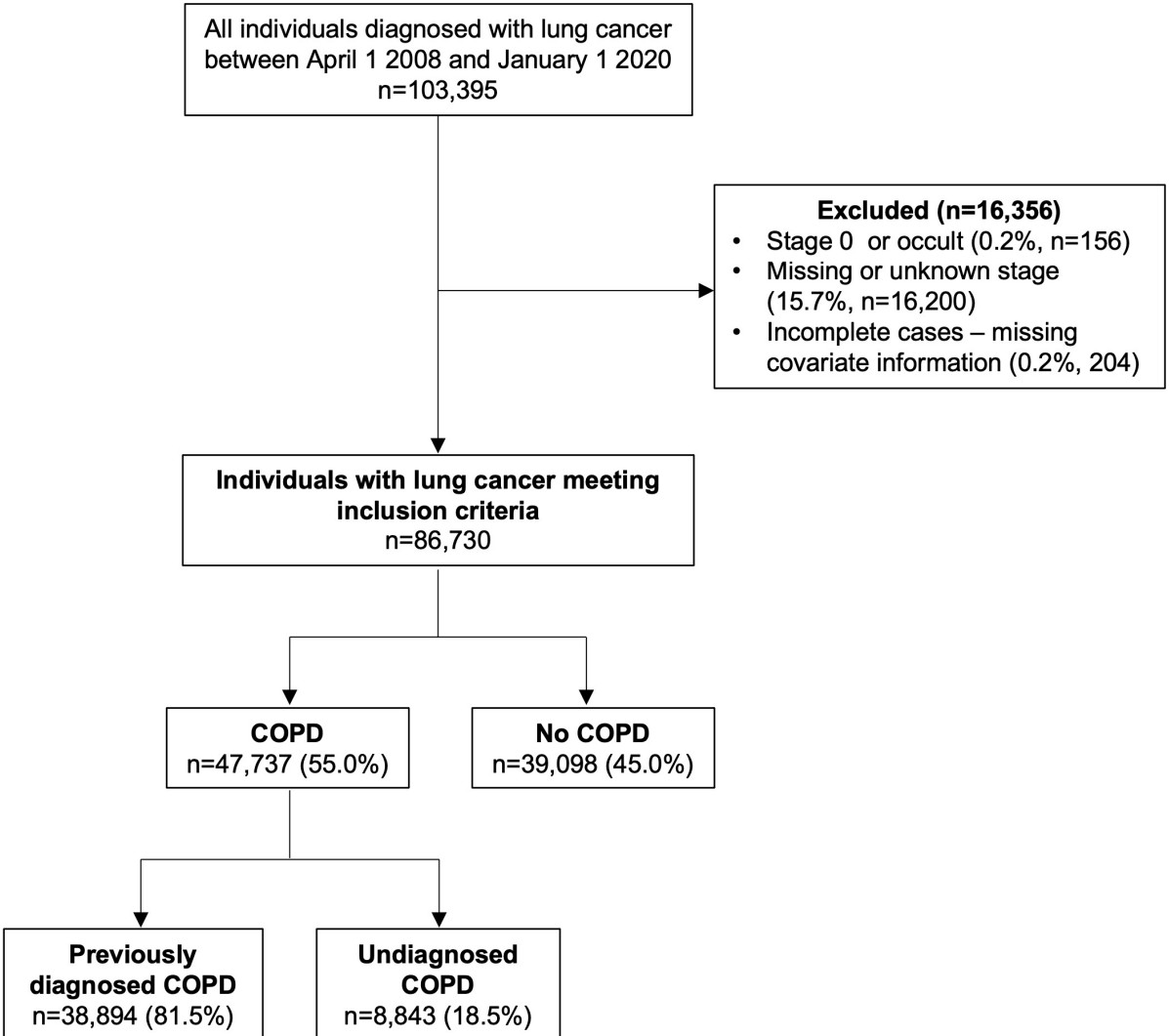

**Figure 1.** Flow diagram depicting the study cohort.

*3.1. COPD and Stage of Lung Cancer*

Most patients (72.1%) were diagnosed with advanced lung cancer (stage III/IV). Advanced-stage diagnosis was highest in individuals without COPD (76.1%) or undiagnosed COPD (72.8%) and lowest in individuals with previously diagnosed COPD (68.0%) (Table 1).

Table 2 depicts the balance in covariates among the COPD exposure groups prior to and after IPW (see Supplementary Tables S2–S4 for the balance in covariates in the subgroup analyses). The logistic regression model results are depicted in Table 3. Compared to people without COPD, people with COPD had 30% lower odds of being diagnosed with lung cancer in the advanced stages (OR = 0.70, 95% CI: 0.68 to 0.72). Findings were consistent regardless of whether COPD was previously diagnosed (OR = 0.68, 95% CI: 0.66 to 0.70) or undiagnosed (OR = 0.77, 95% CI: 0.73 to 0.82). Within the COPD cohort, people with undiagnosed COPD had 18% higher odds of being diagnosed with advanced-stage cancer when compared to people with previously diagnosed COPD (OR = 1.18, 95% CI: 1.12 to 1.24).

**Table 2.** Balance of characteristics prior to and after inverse propensity score weighting (IPW) for individuals with and without COPD.

| | Unweighted | | | Weighted | | |
|---|---|---|---|---|---|---|
| **Characteristics** | COPD | No COPD | | COPD | No COPD | |
| | (*n* = 47,737) | (*n* = 39,098) | SMD | COPD | No COPD | SMD |
| Age, mean (SD) | 71.3 (9.6) | 68.9 (11.3) | 0.229 | 70.4 (9.7) | 70.3 (11.3) | 0.003 |
| Sex, % male | 52.10% | 51.00% | 0.022 | 51.90% | 51.90% | 0 |
| **Rurality/Income Quintile** | | | | | | |
| Rural | 16.3% | 13.8% | 0.072 | 15.2% | 15.1% | 0.003 |
| Urban 1 (lowest) | 22.9% | 18.2% | 0.115 | 20.8% | 20.6% | 0.004 |
| Urban 2 | 19.7% | 18.8% | 0.023 | 19.4% | 19.5% | 0.002 |
| Urban 3 | 15.8% | 17.1% | 0.034 | 16.4% | 16.4% | 0.001 |
| Urban 4 | 13.6% | 16.5% | 0.082 | 14.8% | 14.9% | 0.003 |
| Urban 5 (highest) | 11.6% | 15.6% | 0.117 | 13.5% | 13.5% | 0.003 |
| **Immigration Category** | | | | | | |
| ≤10 years | 0.6% | 2.2% | 0.140 | 1.3% | 1.3% | 0.004 |
| Long-term resident (>10 years) | 2.9% | 6.6% | 0.177 | 4.6% | 4.6% | 0.003 |
| Non-immigrant | 96.5% | 91.2% | 0.224 | 94.2% | 94.1% | 0.005 |
| **Type of Lung Cancer** | | | | | | |
| NSCLC: Adenocarcinoma | 36.7% | 49.4% | 0.256 | 42.5% | 42.9% | 0.008 |
| NSCLC: Squamous cell carcinoma | 21.7% | 13.9% | 0.202 | 18.2% | 18.1% | 0.001 |
| NSCLC: Large cell carcinoma | 2.5% | 2.7% | 0.015 | 2.6% | 2.6% | 0.002 |
| NSCLC: Adenosquamous | 0.6% | 0.6% | 0.006 | 0.6% | 0.6% | 0 |
| NSCLC: Not otherwise specified | 14.1% | 14.2% | 0.005 | 14.1% | 14.0% | 0.003 |
| SCLC | 12.5% | 10.4% | 0.066 | 11.5% | 11.4% | 0.004 |
| Unspecified | 12.0% | 8.8% | 0.104 | 10.5% | 10.3% | 0.006 |
| **Comorbidities** | | | | | | |
| Asthma | 23.1% | 7.9% | 0.412 | 16.1% | 15.4% | 0.021 |
| Congestive heart failure | 16.2% | 7.5% | 0.267 | 12.3% | 12.0% | 0.008 |
| Dementia | 5.0% | 3.7% | 0.064 | 4.4% | 4.40% | 0.002 |
| Diabetes | 28.3% | 24.8% | 0.080 | 26.8% | 26.5% | 0.006 |
| Previous pneumonia | 30.9% | 18.5% | 0.287 | 25.2% | 24.8% | 0.009 |
| Cancer in the previous 5 years | 8.3% | 9.1% | 0.026 | 8.7% | 8.7% | 0 |
| **Rate of Primary Care Visits** | | | | | | |
| ≥5 per year | 47.0% | 34.0% | 0.266 | 41.1% | 40.7% | 0.008 |
| >2 and <5 per year | 35.4% | 38.1% | 0.056 | 36.7% | 37.0% | 0.006 |
| >0 and ≤2 per year | 14.9% | 22.3% | 0.192 | 18.3% | 18.3% | 0.002 |
| 0 | 2.6% | 5.6% | 0.152 | 3.9% | 4.0% | 0.003 |
| Specialist care | 46.9% | 36.4% | 0.214 | 42.2% | 41.9% | 0.006 |

Abbreviations: NSCLC = non-small cell lung cancer, SCLC = small cell lung cancer, SMD = standardized mean difference, SD = standard deviation.

**Table 3.** Association with advanced-stage diagnosis of lung cancer by COPD status.

| | | Odds of Advanced-Stage Lung Cancer Diagnosis Odds Ratio (95% CI) | | | |
|---|---|---|---|---|---|
| | Exposure Groups | Unweighted Univariable Analysis | Weighted Univariable Analysis | Weighted Multivariable Analysis | Weighted Multivariable Analysis Adjusted for Prior Imaging |
| Overall analysis | COPD | 0.70 (0.68 to 0.72) | 0.72 (0.70 to 0.75) | 0.70 (0.68 to 0.72) | 0.77 (0.75 to 0.80) |
| | No COPD | Reference | Reference | Reference | Reference |
| | Prior chest CT scan | – | – | – | 0.36 (0.35 to 0.38) |
| | No prior chest CT scan | | | | Reference |
| Subgroup analysis: Previously diagnosed COPD vs. no COPD | Previously diagnosed COPD | 0.67 (0.65 to 0.69) | 0.70 (0.68 to 0.73) | 0.68 (0.66 to 0.70) | 0.77 (0.75 to 0.80) |
| | No COPD | Reference | Reference | Reference | Reference |
| | Prior chest CT scan | – | – | – | 0.36 (0.35 to 0.38) |
| | No prior chest CT scan | | | | Reference |
| Subgroup analysis: Undiagnosed COPD vs. no COPD | Undiagnosed COPD | 0.84 (0.80 to 0.89) | 0.80 (0.76 to 0.84) | 0.77 (0.73 to 0.82) | 0.77 (0.73 to 0.81) |
| | No COPD | Reference | Reference | Reference | Reference |
| | Prior chest CT scan | – | – | – | 0.31 (0.30 to 0.33) |
| | No prior chest CT scan | | | | Reference |
| Subgroup analysis: Previously diagnosed COPD vs. Undiagnosed COPD | Undiagnosed COPD | 1.26 (1.19 to 1.32) | 1.16 (1.10 to 1.22) | 1.18 (1.12 to 1.24) | 1.05 (1.00 to 1.11) |
| | Previously diagnosed COPD | Reference | Reference | Reference | Reference |
| | Prior chest CT scan | – | – | – | 0.40 (0.39 to 0.42) |
| | No prior chest CT scan | | | | Reference |

Prior chest CT scan refers to individuals who had at least one chest CT scan in the 5 years prior to the diagnostic window. Multivariable analyses were adjusted for variables included in the propensity score estimation: age, sex, rurality and urban income quintile, immigration status, type of lung cancer, comorbidities (asthma, dementia, diabetes, congestive heart failure, a non-lung cancer in the previous 5 years), rate of primary care visits per year in the previous 5 years, and specialist care in the previous 5 years (cardiologist, respirologist, internal medicine).

### 3.2. Impact of Prior Imaging

A total of 22,430 individuals (25.8%) had at least one chest CT scan in the five years prior to the diagnostic window. Individuals with previously diagnosed COPD were more likely to have a chest CT scan during this period (35.8%) compared to individuals without COPD (17.8%, SMD = 0.416). There was no difference in the proportion of patients having chest CT scans during this period among individuals with undiagnosed COPD compared to individuals without COPD (17.6% vs. 17.8%, SMD = 004).

Having a prior CT scan was significantly associated with lower odds of advanced-stage lung cancer (OR = 0.36, 95% CI: 0.35 to 0.38) and slightly attenuated the association between COPD and advanced stage (OR = 0.77, 95% CI: 0.75 to 0.80). The association between undiagnosed COPD and stage of lung cancer diagnosis was unaffected by prior imaging (OR = 0.77, 95% CI: 0.73 to 0.81). After accounting for prior imaging, there was no difference in stage of lung cancer among individuals with undiagnosed COPD compared to individuals with previously diagnosed COPD (OR = 1.05, 95% CI: 1.00 to 1.11).

### 3.3. Other Factors Associated with Stage of Lung Cancer

In the multivariable weighted regression models, many additional covariates were associated with the stage of lung cancer (Figure 2). After adjusting for prior imaging, advanced-stage diagnosis was more likely in males and people with diabetes, congestive heart failure, or previous pneumonia. Living in a high-income urban area or in a rural area was associated with lower odds of advanced-stage lung cancer compared to low-income urban areas. Recent immigrants also had lower odds of advanced-stage lung cancer

compared nonimmigrants. Individuals with dementia also had a marginally lower odds of advanced-stage diagnosis. Of note, a co-diagnosis of asthma was also associated with lower odds of advanced-stage lung cancer (OR = 0.90, 95% CI: 0.86 to 0.94). Higher rates of primary care visits or surveillance for a different (non-lung) cancer in the previous five years were all associated with lower odds of advanced-stage lung cancer. Prior specialist care was also associated with lower odds of advanced-stage cancer (OR = 0.77, 95% CI: 0.75 to 0.80); however, this was attenuated by accounting for prior CT scans (OR = 0.89, 95% CI: 0.86 to 0.92). Important associations between histology and stage were also noted. Individuals with NSCLC or an unspecified type of lung cancer had lower odds of advanced-stage diagnosis compared to individuals with SCLC.

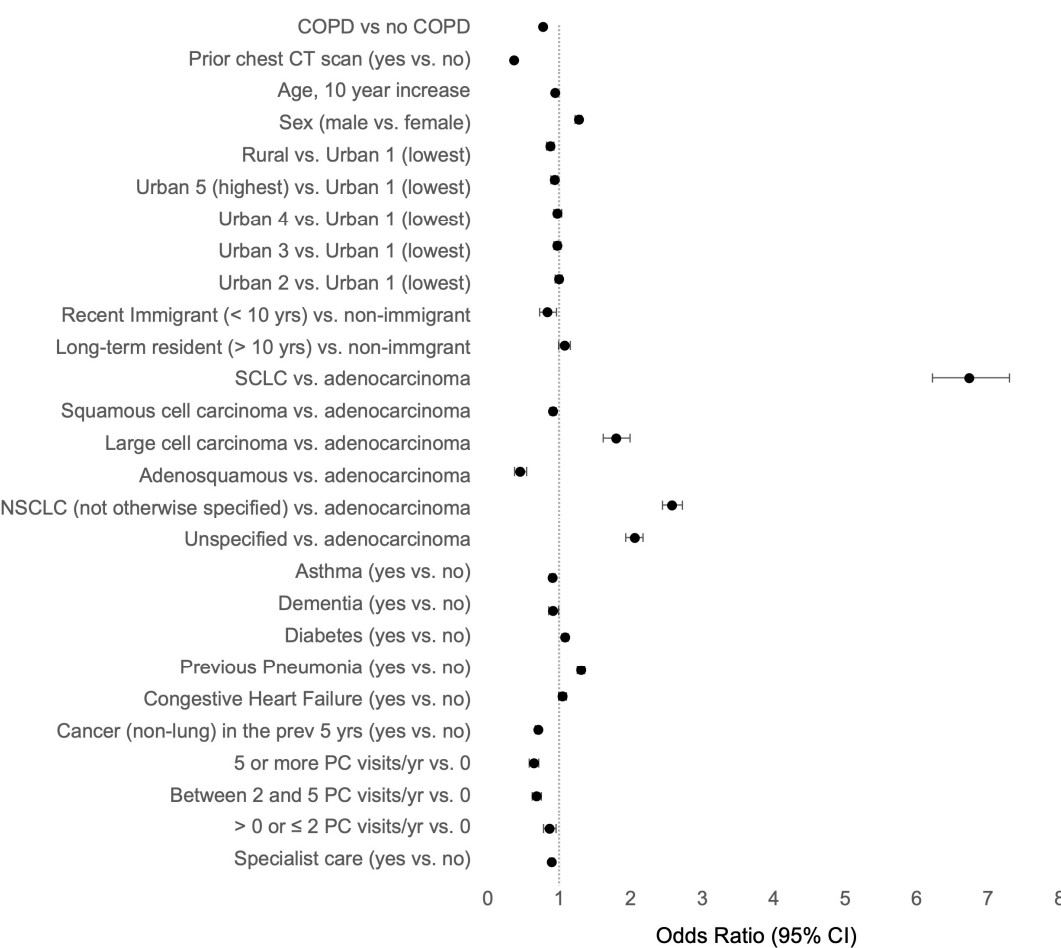

**Figure 2.** Multivariable weighted logistic regression model results depicting the odds of advanced-stage diagnosis. Odds ratios (OR) and 95% confidence intervals (CI) are depicted.

*3.4. Sensitivity Analyses*

The sensitivity analyses results are depicted in Table 4 (see Supplemental Table S5 for balance in the characteristics of individuals with and without COPD for the sensitivity analyses). Findings were consistent when accounting for potential misclassification of COPD by using a definition with higher specificity (OR = 0.64, 95% CI: 0.62 to 0.66). Similar to the main analysis, this finding was also attenuated by adjusting for prior CT scans (OR = 0.74, 95% CI: 0.72 to 0.77).

**Table 4.** Sensitivity analysis for the potential misclassification of COPD.

| Subgroup Analysis | Odds of Advanced-Stage Lung Cancer Diagnosis Odds Ratio (95% CI) | | | |
|---|---|---|---|---|
| | Unweighted Univariable Analysis | Weighted Univariable Analysis | Weighted Multivariable Analysis | Weighted Multivariable Analysis Adjusted for Prior Imaging |
| COPD * | 0.65 (0.63 to 0.67) | 0.66 (0.64 to 0.68) | 0.64 (0.62 to 0.66) | 0.74 (0.72 to 0.77) |
| No COPD | Reference | Reference | Reference | Reference |
| Prior chest CT scan | – | – | – | 0.37 (0.36 to 0.38) |
| No prior chest CT scan | | | | Reference |

* COPD was ascertained using a case definition with higher specificity to account for potential misclassification.

## 4. Discussion

Our population-based study found that most lung cancer patients in Ontario were diagnosed in the advanced stages and more than half had underlying COPD. People with COPD had lower odds of advanced-stage lung cancer. This finding remained even after adjusting for prior imaging and accounting for important confounding variables including income, rurality, established healthcare access, and comorbidities through inverse propensity score weighting. Within the COPD cohort, nearly one in five patients had undiagnosed COPD that was presumably discovered incidentally because of their clinical assessment for lung cancer. In contrast to previously diagnosed COPD, undiagnosed COPD was associated with detection of lung cancer in the advanced stages. These findings highlight the importance of early diagnosis of COPD itself to identify and improve surveillance of people at high risk of developing lung cancer. Our findings also support the need for enhanced partnerships with COPD care providers when caring for lung cancer patients.

Routine imaging to monitor nodules or imaging in response to an exacerbation could lead to incidental findings of lung cancer among individuals with COPD. Our findings suggest that prior imaging only accounted for some of the association between COPD and lower odds of advanced-stage lung cancer diagnosis. We also accounted for prior healthcare access through inverse propensity score weighting. Not surprisingly, we found that prior chest imaging and interactions with the healthcare system were also associated with lower odds of advanced-stage diagnosis. These findings support the importance of healthcare access in the early detection of lung cancer. Our findings are consistent with a previous study by Lofters et al. [31] who showed that higher utilization of primary care was associated with earlier diagnosis of lung cancer. Similar findings have also been noted for colorectal cancer [32]. Additionally, we found that having visited a respirologist, cardiologist, or internal medicine physician was also associated with lower odds of advanced-stage diagnosis. Poor access to primary care or specialists are important barriers preventing early-stage diagnosis [10,33] and should be considered when designing lung cancer screening programs and other interventions which promote early detection. Issues with healthcare access exist even within a country with universal healthcare such as Canada and may play an even greater role in countries where there are additional financial barriers to healthcare. It is important for lung cancer screening programs to target populations that are less likely to seek healthcare, such as males and current or former smokers [11,16,17].

In our study, nearly one in five individuals with COPD had "undiagnosed" disease which was likely found incidentally as part of a lung cancer assessment. A lung cancer screening cohort in the United Kingdom also reported a high prevalence of undiagnosed COPD (67%) with half of these individuals being symptomatic [34]. Earlier diagnosis of COPD itself would allow for more opportunities to intervene with COPD-related treatment, as well as to identify patients at high risk of developing lung cancer who could participate in formal lung cancer screening programs.

In our study, undiagnosed COPD was associated with lower odds of advanced-stage lung cancer when compared to individuals without COPD. One plausible explanation for this association is that people with early-stage lung cancer who are candidates for surgery undergo a substantial clinical assessment which includes spirometry and, thus, incidental findings of COPD may occur. Early-stage diagnosis of lung cancer among patients with COPD could also be explained by the overlap in symptoms. We were unable to account for symptoms in our study; however, previous research has shown that patients with lung-related symptoms such as cough, shortness of breath, chest pain, or hemoptysis had shorter median times to diagnosis than people with non-lung-related symptoms or vague symptoms such as fatigue [13]. Even people with undiagnosed COPD can be symptomatic [35] despite generally having less severe airflow obstruction than individuals with diagnosed COPD [36]. A recent study by Dai et al. (2021) [37] found that symptom prevalence among stage I lung cancer patients was similar regardless of whether COPD was "incidental" (diagnosed within six months prior to lung cancer) or previously diagnosed [37]. They also found that lung cancer patients with comorbid COPD were more likely to present with respiratory symptoms compared to lung cancer patients without COPD [37].

Similar to COPD, we found that asthma was also associated with lower odds of advanced-stage diagnosis. In contrast, after accounting for confounding factors, nonrespiratory comorbidities such as congestive heart failure and diabetes were associated with increased odds of advanced-stage diagnosis. Prior research has shown that diabetes is associated with advanced-stage diagnosis of breast cancer [38], with undiagnosed diabetes placing patients at an even higher risk of advanced-stage diagnosis [39]. This further supports the need to consider specific comorbidities in cancer research.

Our study has many notable strengths. First, we utilized a validated algorithm to identify COPD [20] and accounted for undiagnosed COPD which is often overlooked. Second, our sample size was large due to the use of linked cancer registries and health administrative datasets. This allowed us to account for many confounding variables and improve the generalizability of our results. Lastly, our strong methodological approach using IPW allowed us to account for inherent differences in individuals with and without COPD, effectively minimizing confounding bias.

However, our study is not without limitations. Although the Ontario Cancer Registry (OCR) is population-based, it differs from the US-SEER registries which are prospectively ascertained. A significant proportion (16%) of individuals with lung cancer did not have stage information recorded in the OCR. In general, patients with missing stage were older, had more comorbidity, and were less likely to have the type of lung cancer specified or confirmed via histology/cytology. Our databases are also devoid of spirometry results and we could not ascertain the severity of COPD according to GOLD criteria [40]. Severity of COPD has important implications for lung cancer screening and survival [5,41,42] and could influence the stage at which patients present. Misclassification of COPD is also a possibility; however, we accounted for this by repeating our analysis with a stricter case definition with higher specificity, albeit a lower sensitivity and found similar results. There may also be unmeasured confounding as we do not have access to information on other factors that could be associated with the stage of lung cancer diagnosis (e.g., family history of lung cancer, mutational status, or smoking history). Current or former smokers may be at risk of advanced-stage lung cancer diagnosis as a result of avoiding seeking care due to self-blame or perceived stigma from healthcare providers for smoking [16,17]. Lastly, we were unable to ascertain whether lung cancer was diagnosed through informal or opportunistic screening. We did, however, adjust for prior chest CT scans in an attempt to account for informal screening or monitoring of lung nodules.

Our results have important clinical implications. Although our findings determined that a diagnosis of COPD is associated with lower odds of advanced-stage lung cancer at diagnosis, outcomes for patients with COPD are historically worse, especially among patients diagnosed with early-stage disease [5,6]. There is an urgent need to optimize care

and improve survival rates for early-stage lung cancer patients with COPD. Future studies should address whether management of COPD itself can improve outcomes for lung cancer patients with comorbid disease, as well as assess the effectiveness of lung cancer treatments among this population.

Lastly, despite the positive association between early-stage diagnosis and COPD, most patients with lung cancer in our study were diagnosed with advanced-stage disease. Formal lung cancer screening programs and awareness campaigns are important steps to promote earlier detection [33] and reduce mortality [43]. There is also a unique opportunity to adapt these programs and promote earlier detection of COPD as well, given the similar target population.

## 5. Conclusions

To our knowledge, this is the first study to indicate that, in a real-world setting and the absence of formal screening programs, COPD is associated with lower odds of diagnosing lung cancer in the advanced stages. Lung cancer diagnostics and early detection programs may benefit from enhanced partnership with COPD healthcare providers, and interventions should strive to promote earlier detection of both diseases.

**Supplementary Materials:** The following supporting information can be downloaded at: https://www.mdpi.com/article/10.3390/curroncol30070471/s1, Table S1: Lung cancer morphology codes; Tables S2–S5: Balance of characteristics for subgroup and sensitivity analyses prior to and after inverse propensity score weighting.

**Author Contributions:** All authors contributed to the study conceptualization, methodology, and provided oversight to the analysis plan. Author S.J.B. analyzed the data. All authors interpreted the data. S.J.B. drafted the manuscript. All authors have read and agreed to the published version of the manuscript.

**Funding:** This study received funding from the University of Toronto, Department of Respiratory Medicine, Godfrey S. Pettit Block Term Grant and was supported by the Canadian Institutes of Health Research. Author A.S.G. receives funding from the Ontario Ministry of Health (MOH) and Ministry of Long-term Care (MLTC). Authors A.V.L. and L.P. receive clinician scientist funding from the Ontario MOH via the Ontario Association of Radiation Oncologists. This study was also supported by ICES which is funded by an annual grant from the Ontario MOH and MLTC. This document used data adapted from the Statistics Canada Postal Code^OM^ Conversion File, which is based on data licensed from Canada Post Corporation, and/or data adapted from the Ontario Ministry of Health Postal Code Conversion File, which contains data copied under license from ©Canada Post Corporation and Statistics Canada. Parts of this material are based on data and/or information compiled and provided by: Ontario Health, the Canadian Institute for Health Information (CIHI) and the Ontario Ministry of Health, and Immigration, Refugees and Citizenship Canada (IRCC) current to May 2017. The analyses, conclusions, opinions, and statements expressed herein are solely those of the authors and do not reflect those of the funding or data sources; no endorsement is intended or should be inferred.

**Institutional Review Board Statement:** The study was conducted in accordance with the Declaration of Helsinki, and approved by the University of Toronto Health Sciences Research Ethics Board (protocol # 39487, date of approval: 9 July 2020).

**Informed Consent Statement:** Patient consent was waived. ICES is a prescribed entity under the Ontario's Personal Health Information Protection Act (PHIPA). Section 45 of PHIPA authorizes ICES to collect personal health information, without consent, for the purpose of analysis or compiling statistical information with respect to the management, evaluation, or monitoring of the allocation of resources to or planning for all or part of the heath system. The use of the data in this project is authorized under Section 45 of PHIPA and approved by ICES' Privacy and Legal Office.

**Data Availability Statement:** The dataset from this study is held securely in coded form at ICES. While legal data sharing agreements between ICES and data providers (e.g., healthcare organizations and government) prohibit ICES from making the dataset publicly available, access may be granted to those who meet prespecified criteria for confidential access, available at https://www.ices.on.ca/services-for-researchers/submit-a-request/ (accessed on 4 July 2023) (email: das@ices.on.ca). The

full dataset creation plan and underlying analytic code are available from the authors upon request, understanding that the computer programs may rely upon coding templates or macros that are unique to ICES and are therefore either inaccessible or may require modification.

**Acknowledgments:** The authors would like to thank Ruth Croxford, Senior Analyst at ICES for her expertise and assistance.

**Conflicts of Interest:** AVL has received speaking fees from AstraZeneca and is a member of the advisory board. Authors S.J.B., R.S., L.P., D.B. and A.S.G. declare no conflict of interest.

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
