# Peer review of "Association between COPD and Stage of Lung Cancer Diagnosis: A Population-Based Study"

_curroncol, doi:10.3390/curroncol30070471_

Round 1
Reviewer 1 Report
The authors compare people with diagnosed lung cancer and determine the relation between COPD and lung cancer, noting those with COPD were diagnosed at an earlier stage ( perhaps due to more interactions with health care/prior imaging). The limitations are well described (i.e. - this starts with lung cancer diagnosis so really does not examine the rellation between COPD and lung cancer, per se)
Comments:
Figure 2 is the centerpiece of the paper and could be improved by
1. adding a line for Undiagnosed COPD ( i.e.- compare Diagnosed to No COPD and Undiagnosed to No COPD)
2. Do age in 10 year interval rather than 1 year
3. Have Adenocarcinoma as the reference category instead of small cell ( which are all advanced)
In Table 2 the second adenocarcinoma line should be adenosquamous
I found Table 3 a bit confusing. It might be better to have Previous COPD, Undiagnosed COPD, and NO COPD ( reference) presented in one grouping and then do the same, but stratified by prior CT chest scan in another grouping.
Reviewer 2 Report
Comments to authors
Authors evaluated the impact of COPD on the stage of lung cancer diagnosis. In terms of methodology, the statistical analyses and article format are well organized. The article provides novel information about COPD is associated with lower odds of diagnosing lung cancer in the advanced stages that is rarely reported in the literature. It can promote earlier detection of both diseases. Although this manuscript has several problems that need to be revised, the study is still clinically meaningful.
Here are some issues that needs to be addressed.
Specific comment
1、The purpose of the abstract(line 14) seems to determine the association between COPD and stage of lung cancer, but the purpose of the introduction(line 62) was to determine the association between whether previously diagnosed COPD and stage of lung cancer. These are two different purposes, which makes me confused. I think the author needs to unify the purpose of abstract and introduction.
2、Introduction
Line 43: “Lung cancer may be detected …routine assessment of respiratory symptoms. ” requires a reference.
3、2.3. Study Population
Line 89: Why was the individuals with stage 0 being excluded?
4、2.4. Primary Outcome
Line 93-96: What is the “the ‘Best Stage’ definition from the OCR” ? What if the same patient has different clinical and pathological stages? Why not use a uniform method of staging?
5、2.5. Exposures
Line104-106: The definition of ‘previously diagnosed COPD’ is > 90 days prior to the diagnosis of lung cancer. Why did authors choose 90 days?
6、2.6 Covariates
Smoking is associate with COPD and lung cancer. Why did the authors not include smoking status of patients?Why did author not accounting for smoking as one of the variables which are associated with lung cancer diagnostic stage?
7、Table1: Maybe I am getting confused here -The “Type of Lung Cancer” shown in Table 1 should be based on histological/ Cytology, so the percentage of “Type of Lung Cancer” except “Unspecified” should be the same as the “Histology/Cytology” of “Method of Confirmation”. For example, the total percentage of “Type of Lung Cancer” except “Unspecified” in the column of “Total COPD” is 88.1%, but the “Histology/Cytology” percentage of “Method of Confirmation” is 87.1%. I could not understand that these two percentages are not equal, as was the other columns.
8. Some references are too old. I suggest more quoting authoritative and recent research results.
Round 2
Reviewer 1 Report
I still think Figure 2 would be better with Adenocarcinoma as the reference ( and the results presented on a log scale) but will defer to the editors on that.
Other than that- I'm fine with the revisions.
Author Response
We have revised figure 2 to have 'adenocarcinoma' as the reference group for the type of lung cancer. As evident in the revised figure, the odds ratio for SCLC vs. adenocarcinoma is much larger than the odds ratios for the other variables and this skews the x-axis so that the odds ratios < 1 are minimized. We acknowledge the reviewer's comment that presenting this figure on a log-scale would fix this issue, however a log-scale would not be easily interpretable and also would not coincide with the odds ratios presented in table 3. However, if the editors do feel that presenting this figure on a log-scale would be more appropriate, we can revise this figure.